# Long-Term Protein Restriction Modulates Lipid Metabolism in White Adipose Tissues and Alters Colonic Microbiota of Shaziling Pigs

**DOI:** 10.3390/ani12212944

**Published:** 2022-10-26

**Authors:** Jie Zheng, Yehui Duan, Changbing Zheng, Jiayi Yu, Fengna Li, Qiuping Guo, Yulong Yin

**Affiliations:** 1CAS Key Laboratory of Agro-Ecological Processes in Subtropical Region, Hunan Provincial Key Laboratory of Animal Nutritional Physiology and Metabolic Process, National Engineering Laboratory for Pollution Control and Waste Utilization in Livestock and Poultry Production, Institute of Subtropical Agriculture, Chinese Academy of Sciences, Changsha 410125, China; 2University of Chinese Academy of Sciences, Beijing 100039, China

**Keywords:** protein restriction, lipid metabolism, obesity, gut microbiota, Shaziling pigs

## Abstract

**Simple Summary:**

The effects of prolonged protein deprivation on lipid metabolism in white adipose tissues and the gut microbiota of Shaziling pigs remain inadequately characterized. As an ideal obese model, the Shaziling pig is instrumental in studying obesity. This study was developed to examine the changes in lipid metabolism and the gut microbiota ecosystem induced by prolonged protein deficiency of Shaziling pigs and to explore the beneficial effects of low-protein diets for obesity management. After a 24-week trial, increased FABP4 mRNA abundance and PPARγ protein expression as well as decreased C/EBPα protein expression in dorsal subcutaneous adipose tissue and perirenal adipose tissue were achieved through a 20% reduction in the dietary protein level. At the same time, increased/decreased phosphorylation of AMPKα/mTOR were observed. Furthermore, reducing the protein level by 20% increased the relative abundance of the *Lachnospiraceae XPB1014* group in the colon at the genus level. In summary, our findings indicated that long-term reduction in dietary crude protein by 20% positively impacted the lipid metabolism of Shaziling pigs.

**Abstract:**

Obesity is a matter of concern to the public. Abundant evidence has been accumulated that nutritional intervention is a promising strategy to address this health issue. The objective of this study is to investigate alterations in the lipid metabolism in white adipose tissues and the gut microbiota of Shaziling pigs challenged by long-term protein restriction. Results showed that compared with the control group, reducing the protein level by 20% (−20%) increased the mRNA abundance of FABP4 in white adipose tissues (*p* < 0.05). This occurred in conjunction with increases in PPARγ protein expression. Conversely, the protein expression of C/EBPα was reduced in the −20% group (*p* < 0.05). Moreover, the −20% group had increased/decreased phosphorylation of AMPKα/mTOR, respectively (*p* < 0.05). As for the colonic gut microbiota, a 20% reduction in the protein level led to increased *Lachnospiraceae XPB1014* group abundance at the genus level (*p* < 0.01). Collectively, these results indicated that a 20% protein reduction could modulate lipid metabolism and alter the colonic microbiota of Shaziling pigs, an approach which might be translated into a treatment for obesity.

## 1. Introduction

Obesity has reached epidemic levels across the globe, and nutritional manipulations for reducing adipose tissue mass are under intensive scientific scrutiny. Since pigs share significant similarities in anatomy and physiology with humans, they are recognized as an appropriate model for researching lipid metabolism and obesity [1]. Indeed, we previously observed that the piglets’ mRNA and protein levels of lipogenic-related genes were affected by low-protein diets in a depot-specific manner [2]. Additionally, the pig’s gut harbors a tremendous number of bacteria that interact with the host and play a crucial role in regulating lipid metabolism [3]. Experiments on weaned pigs suggested that grape seed proanthocyanidin intake reduced the adipocyte sizes of subcutaneous and visceral adipose tissues via altering the gut microbiota [4]. Similarly, our previous study found that growing pigs consuming β-hydroxy-β-methylbutyrate supplementation had decreased fat mass [5], and this improvement might be associated with increases in the relative abundance of *Bacteroidetes* [6].

Numerous papers have shown that varying dietary protein levels exerts effects on the intestinal microflora. For example, low-protein diets (10%) for finishing pigs significantly decreased the abundance of *Halanaerobium* and *Butyricicoccusat* at the genus level in the colonic digesta [7]. In growing pigs, as the dietary protein level was reduced from 18% to 14%, the relative abundance of *Tenericutes* in ileal digesta decreased, whereas that of *Cyanobacteria* increased [8]. Piglets fed a low-protein diet (14%) had reduced numbers of *Firmicutes* and *Clostridium cluster IV* in the caecal digesta compared with the control group (20%) [9]. Moreover, investigators have demonstrated that a periodized low-protein diet reduced the abundance of Desulfovibrionaceae, which was associated with the loss of fat mass [10]. This suggested that the obesity-protection effect of protein restriction might be mediated by changes in the gut microbiota. Considering the regulatory implications of low-protein diets in fat deposit and the possible benefits to combat obesity, the effects of protein restriction on the lipid metabolism signaling and the gut microbiota deserve careful research.

Although previous studies have described alterations in lipid metabolism and the gut microbiota during certain growth periods of pigs, relevant information about pigs with long-term and consecutive protein restriction requires further investigation. Moreover, the majority of existing studies were carried out on commercial breeds. Compared with commercial breeds, the Shaziling pig exhibits a fatter phenotype, which makes it a better subject for studying obesity. Here, we employed the Shaziling pig, a native fat pig breed, to explore the regulatory role of long-term protein restriction in some key signaling molecules related to lipid metabolism in white adipose tissues and the gut microbiota. This study is expected to lay the foundation for reducing fat accumulation of pigs and provide some enlightenment for tackling obesity as well.

## 2. Materials and Methods

### 2.1. Animals and Diets

The experiment was approved by the Animal Care Committee of the Institute of Subtropical Agriculture, Chinese Academy of Sciences, and the ethic approval number is ISA-2020-023.

Animals and diets were specifically recorded in our preceding study [11]. In brief, forty Shaziling piglets were randomly distributed into five treatment groups and supplied with diets containing different crude protein (CP) levels (abbreviated as +20%, +10%, 0 (control), −10%, and −20%, respectively). The control group was fed a normal protein diet according to the nutrition requirements of the Shaziling pig (Appendix A).

### 2.2. Sample Collection

Blood samples (5 mL) from the overnight fasting pigs were obtained by jugular vein puncture using anticoagulant-free vacuum tubes and centrifuged at 3000 *g* at 4 °C for 15 min. Then, the serum was collected and stored at −80 °C until analysis. After blood sampling, pigs were electrically stunned (250 V, 0.5 A, 5–6 s), exsanguinated, and eviscerated. The colon was quickly separated, and the colonic contents were collected from a region 10 cm posterior to the ileocecal valve into sterile tubes. Meanwhile, samples of dorsal subcutaneous adipose tissue (DSA) and perirenal adipose tissue (PRA) were excised from the right side of the carcass. Samples of colonic contents and adipose tissue were immediately stored at −80 °C for subsequent tests.

### 2.3. Blood Chemical Parameters of Lipid Metabolism

The concentrations of triglyceride (TG), cholesterol (CHOL), low-density lipoprotein-cholesterol (LDL-C), and high-density lipoprotein-cholesterol (HDL-C) in serum were measured using the automatic biochemical analyzer (Cobas c311, Roche, Switzerland) and commercial kits (Sigma Aldrich Trading Co., Ltd., Shanghai, China).

### 2.4. Real-Time Quantitative PCR Analysis

Total RNA from DSA and PRA was extracted using AG RNAex Pro Reagent (Accurate Biotechnology (Hunan), Co., Ltd., China) according to the manufacturer’s instructions. The purity of the total RNA was verified using a NanoDrop^®^ ND-2000 spectrophotometer (Thermo Scientific Inc., Waltham, MA, USA). The removal of gDNA in the total RNA and reverse transcription were accomplished by using Evo M-MLV RT Kit with gDNA clean for qPCR (Accurate Biotechnology (Hunan), Co., Ltd., China). Real-time PCR was conducted with an ABI 7900 PCR system (ABI Biotechnology, Eldersburg, MD, USA). The total reaction system (10 μL) included 2 μL cDNA (5 ng/μL), 5 μL SYBR Green Pro Taq HS Premix (Accurate Biotechnology (Hunan), Co., Ltd., China), 0.2 μL of each primer (10 μM) and 2.6 μL of RNase-free water. The PCR protocol was: incubation for 10 min at 95 °C, followed by 40 cycles of denaturation for 15 s at 95 °C, and annealing and extension for 60 s at 56 °C to 64 °C. All the samples were measured in duplicate. The primers were designed using the Oligo 6.0 software (Table 1). Relative expression of target genes was calculated by the 2^−ΔΔCt^ method [12].

### 2.5. Western Blotting Analysis

An adipose tissue sample (approximately 0.05 g) was fully homogenized with 500 μL RIPA lysate and placed on ice for lysis. The mixture was centrifuged at 12,000 rpm for 15 min at 4 °C and the supernatant was transferred to another 1.5 mL centrifuge tube. The protein concentration was measured by using BCA Protein Assay Kit (meilunbio, Dalian, China), and 20 μg of the total protein was added into SDS—PAGE gel. Next, targeted proteins were separated by electrophoresis and transferred onto nitrocellulose filter membranes. Membranes were sealed with 5% skim milk powder for 90 min at room temperature and then put aside overnight at 4 °C. The primary antibodies were as follows: CCAAT/enhancer-binding protein alpha (C/EBPα, Proteintech, Chicago, IL, USA, 1:1000), peroxisome proliferator-activated receptor gamma (PPARγ, Proteintech, Chicago, IL, USA, 1:1000), phosphorylated (p)-AMP-activated protein kinase AMPK (p-AMPK, Proteintech, Chicago, IL, USA, 1:1000), phosphorylated (p)-Mammalian target of rapamycin (p-mTOR, CST, Boston, MA, USA, 1:1000), and GAPDH (Proteintech, Chicago, IL, USA, 1:5000). The secondary antibody was HRP goat anti-rabbit IgG (Proteintech, Chicago, IL, USA, 1:6000). The protein bands were visualized by a chemiluminescent reagent (Pierce, Rockford, IL, USA) with a ChemiDoc XRS system (Bio-Rad, Philadelphia, PA, USA). The resulting signals were quantified using Alpha Imager 2200 software (Alpha Innotech Corp., San Leandro, CA, USA), and the data were normalized to the inner control GAPDH (Proteintech, Chicago, IL, USA).

### 2.6. Microbiota Analysis

#### 2.6.1. DNA Extraction and PCR Amplification

Total genomic DNA of the microbiota was extracted from colonic content samples using the E.Z.N.A.^®^ Soil DNA Kit (Omega Bio-tek, Norcross, GA, USA). The quality and concentration of DNA were determined by 1.0% agarose gel electrophoresis and a NanoDrop^®^ ND-2000 spectrophotometer (Thermo Scientific Inc., Waltham, MA, USA). The hypervariable regions V3-V4 of the bacterial 16S rRNA gene were amplified with primer pairs 338F (5′-ACTCCTACGGGAGGCAGCAG-3′) and 806R(5′-GGACTACHVGGGTWTCTAAT-3′) [13] by an ABI GeneAmp^®^ 9700 PCR thermocycler (ABI, CA, USA). The PCR reaction mixture contained 4 μL 5 × Fast Pfu buffer, 2 μL 2.5 mM dNTPs, 0.8 μL of each primer (5 μM), 0.4 μL Fast Pfu polymerase, 0.2 μL BSA, 10 ng of template DNA, and ddH_2_O to a final volume of 20 µL. PCR amplification cycling conditions were as follows: initial denaturation at 95 °C (3 min), 27 cycles of denaturing at 95 °C (30 s), annealing at 55 °C (30 s) and extension at 72 °C (45 s), single extension at 72 °C (10 min), and the end at 4 °C. All samples were measured in triplicate. The AxyPrep DNA Gel Extraction Kit (Axygen Biosciences, Union City, CA, USA) was used for extracting and purifying the PCR product, which was then quantified using Quantus™ Fluorometer (Promega, Madison, Wisconson, USA).

#### 2.6.2. Illumina MiSeq Sequencing

Purified amplicons were gathered in equimolar amounts and paired-end sequenced on an Illumina MiSeq PE300 platform (Illumina, San Diego, CA, USA) according to the standard protocols by Majorbio Bio-Pharm Technology Co., Ltd. (Shanghai, China).

#### 2.6.3. Data Processing and Statistical Analysis

Raw FASTQ files were de-multiplexed using an in-house perl script, and then quality-filtered by fastp version 0.19.6 [14] and merged by FLASH version 1.2.7 [15]. To minimize the effects of sequencing depth on the α and β diversity measures, the number of 16S rRNA gene sequences from each sample was rarefied to the minimum number of gene sequences. The taxonomy of each OTU representative sequence was analyzed by RDP Classifier version 2.2 [16] against the 16S rRNA gene database using a confidence threshold of 0.7.

Bioinformatic analysis of the gut microbiota was carried out using the Majorbio Cloud platform (https://cloud.majorbio.com, accessed on 11 July 2022). α-diversity indices including Shannon index and Simpson index were calculated with Mothur v1.30.1 [17]. The similarity among the microbial communities in different samples was determined by non-metric multidimensional scaling (NMDS) based on Bray–Curtis dissimilarity.

### 2.7. Statistical Analyses

Statistical analyses for the blood chemical parameters and the expression of genes and proteins were conducted by SAS 8.2 software (Institute, Inc., Cary, NC, USA), using either the one-way ANOVA procedure or the Kruskal–Wallis test [18]. Results were presented as means ± standard error of the mean (SEM). *p* < 0.05 was recognized as the level of statistical significance.

## 3. Results

### 3.1. Blood Chemical Parameters of Lipid Metabolism

As shown in Table 2, the concentrations of TG, CHOL, LDL-C, and HDL-C did not differ among the dietary treatments. Moreover, we found no linear or quadratic effects for these parameters.

### 3.2. The mRNA Expression of Genes Related to Lipid Metabolism in Adipose Tissues

As revealed in Figure 1 and Figure 2, the mRNA abundance of genes involved in lipogenesis (acetyl-CoA carboxylase, ACC; fatty acid synthase, FAS; and lipoprotein lipase, LPL), lipolysis (hormone-sensitive lipase, HSL), and lipid transport (fatty acid transport protein, FATP1 and fatty acid binding protein 4, FABP4) were measured in DSA and PRA. In DSA (Figure 1), the low-protein diets were not effective in regulating the mRNA expression level of FAS in comparison with the control diets (*p* > 0.05). The gene expression level of FABP4 in the −10% group was similar to that of the control group (*p* > 0.05). However, the −20% protein diets significantly elevated the gene expression level of FABP4 compared with the control group (*p* < 0.05). There were no significant effects in the gene expression of ACC, LPL, HSL, and FATP1 among the experimental treatments (*p* > 0.05). In PRA (Figure 2), no significant differences in the gene expression levels of HSL and FATP1 were observed among the control, −10%, and −20% groups (*p* > 0.05). Compared with the control group, the gene abundance of FABP4 was little affected by the −10% protein diet (*p* > 0.05), but it was markedly higher in the −20% group (*p* < 0.05). The gene expression of ACC, FAS, and LPL did not differ between five groups (*p* > 0.05).

### 3.3. The Protein Expression of Key Molecules Related to Lipid Metabolism in Adipose Tissues

In DSA (Figure 3 and Appendix A), decreased levels of protein linearly reduced the relative protein expression of C/EBPα, with the lowest value in the −20% group (*p* < 0.01). In contrast, the relative protein expression of PPARγ linearly increased as dietary protein levels decreased and achieved the maximum level in the −20% group (*p* < 0.01). Dietary protein reduction also led to linear increases in the relative protein expression of p-AMPK (*p* < 0.01), resulting in the highest point in the −20% group. As protein levels decreased, a linear decrease was observed in the relative protein expression of p-mTOR, and the −20% group exhibited the minimum value (*p* < 0.01). Interestingly, similar observations were found in PRA (Figure 4 and Appendix A), which suggested that there is a close resemblance of lipid metabolisms between DSA and PRA under the condition of low protein availability.

### 3.4. The Gut Microbiota Composition in Colon

As is shown in Figure 5A,B, the reduced Shannon index (*p* < 0.01) and increased Simpson index (*p* < 0.05) in the −20% group indicated a lower α-diversity of the microflora in comparison to the control group. However, the area of each group in the NMDS plot was cross-distributed at family level (Figure 5C), suggesting little differences between all groups in β-diversity (*p* > 0.05).

At the genus level, the relative abundances of the top 10 colonic microbiota of pigs fed different levels of dietary protein are listed in the Table 3. Decreasing the dietary protein levels gave rise to linear increases in the relative abundance of *Unclassified f Lachnospiraceae* (*p* < 0.01), and the highest value was observed in the −10% group. The relative abundance of *Lachnospiraceae XPB1014* group showed a quadratic response to varied protein levels (*p* < 0.01) and reached the maximal value in the −20% group. Moreover, a 20% or 10% reduction in the protein level raised the relative abundance of *Christensenellaceae R-7* group compared with the control group, although the magnitude of the increases was marginal.

## 4. Discussion

Since lipid metabolism is a complicated and integrated process of events involving molecular changes, we first measured the expression of several genes relating to this system in adipose tissues. It is well-established that FABP4 is a cytosolic protein responsible for lipid trafficking in adipocytes. Furthermore, FABP4 can form a complex with HSL on an N-terminal 300-amino acid region of the lipase [19]. As a result, the hydrolytic activity of HSL is increased because FABP4 facilitates the delivery of fatty acids and attenuates the feedback inhibition of released products [20,21]. Thus, adipose lipolysis can be contributed by the protein–protein interaction of FABP4 with HSL. Several findings have converged to demonstrate a driving force of FABP4 in lipolysis. For instance, targeted disruption of the FABP4 gene in lean mice caused diminished lipolytic efficiency and impaired fatty acid efflux [22]. An additional high-fat feeding study reported that compared with C57Bl/6J mice, FABP4-KO mice had reduced lipolysis and enhanced lipogenesis, while the phenotypes of mice with overexpressed FABP4 gene were quite the opposite [23]. In the present study, we found that in both DSA and PRA, the mRNA expression of FABP4 was noticeably higher in the −20% group relative to the control group. Although we did not observe a significant difference in the HSL mRNA abundance between the control group and the −20% group in selected adipose tissues, given the physical association of FABP4/HSL, we speculated that the elevation of FABP4 mRNA levels might accelerate fatty acid movement and boost HSL activity, thereby stimulating lipolysis and preventing fat deposition.

Adipogenesis is greatly influenced by transcriptional signals, among which C/EBPα and PPARγ act as master regulators [24]. It was anticipated that the protein expression of C/EBPα and PPARγ would display a similar change trend. However, the protein expression patterns of these two transcriptional factors were strikingly different in selected adipose tissues under our experimental conditions. On the one hand, the protein levels of C/EBPα were linearly reduced by low-protein feeding, implying blunted adipocyte maturation during terminal white adipogenesis [25]. On the other hand, the variations of PPARγ levels were contrary to those of C/EBPα. So, it is reasonable to assume that the transcriptional regulation of adipogenesis is not just a simple program, but rather a much more complicated network operating in a context- and cell type-dependent manner. Intriguingly, apart from the essential action of promoting adipogenesis, PPARγ also induces the activity of FABP4 promoters [26]. Therefore, the upregulated FABP4 mRNA abundance in the −20% group might be a secondary effect of the increased expression of PPARγ. In addition, we detected AMPK/mTOR signaling to evaluate its involvement in lipid metabolism. mTOR serves pivotal functions in adipocyte growth and metabolism, including promoting adipogenesis, enhancing lipogenesis, and suppressing lipolysis [27]. Furthermore, as an upstream regulator of mTOR, AMPK negatively controls the induction of mTOR. Our previous study proposed that the activation of AMPK and the suppression of mTOR in adipose tissues may contribute to the reduction of total fat weight [28]. Consistently, results in the present study showed the enhancement of p-AMPK and the downregulation of p-mTOR, indicating the dominance of catabolism in adipose tissues when pigs were subjected to limited protein intake. Hence, we believed that low-protein diets could favor a fat-lowering effect via the activated AMPK and the depressed mTOR, with the −20% group being the foremost. Altogether, information gained from these observations highlighted that low-protein feeding may impede adipogenesis and provoke lipolysis, especially with a 20% reduction.

A compelling set of associations between the host diets, the gut microbiota, and lipid metabolism has emerged. In growing pigs, the relative abundance of *Lachnospiraceae XPB1014* group was negatively associated with body fat weight [29]. Moreover, a decreased *Christensenellaceae R-7* group was regarded as a contributor to high fructose corn syrup-induced obesity, and a positive association was found between changes in body weight and *Christensenellaceae R-7* group [30,31]. At the genus level, the increased relative abundances of the *Lachnospiraceae XPB1014* group and the *Christensenellaceae R-7* group were achieved in the −10% group and the −20% group. Thus, low-protein diets are deduced to affect the gut microbiota environment and regulate lipid metabolism in a positive way. However, the specific mechanism is not fully understood, requiring systemic studies to make further insights into this topic.

## 5. Conclusions

Taken together, the above-mentioned findings suggested that Shaziling pigs could adjust their lipid metabolism in white adipose tissues and the colonic microbiota in response to long-term protein restriction. Reducing the protein level by 20% resulted in the increased expression of FABP4 mRNA and PPARγ proteins and activated AMPK/mTOR signaling in both DSA and PRA. Furthermore, an increased abundance of the colonic *Lachnospiraceae XPB1014* group was observed. Such changes in the lipid metabolism network and the relative abundance of microbiota indicated beneficial effects on fat reduction, which may lend credence to the practical use of low-protein diets for the management of obesity.

## Figures and Tables

**Figure 1 animals-12-02944-f001:**
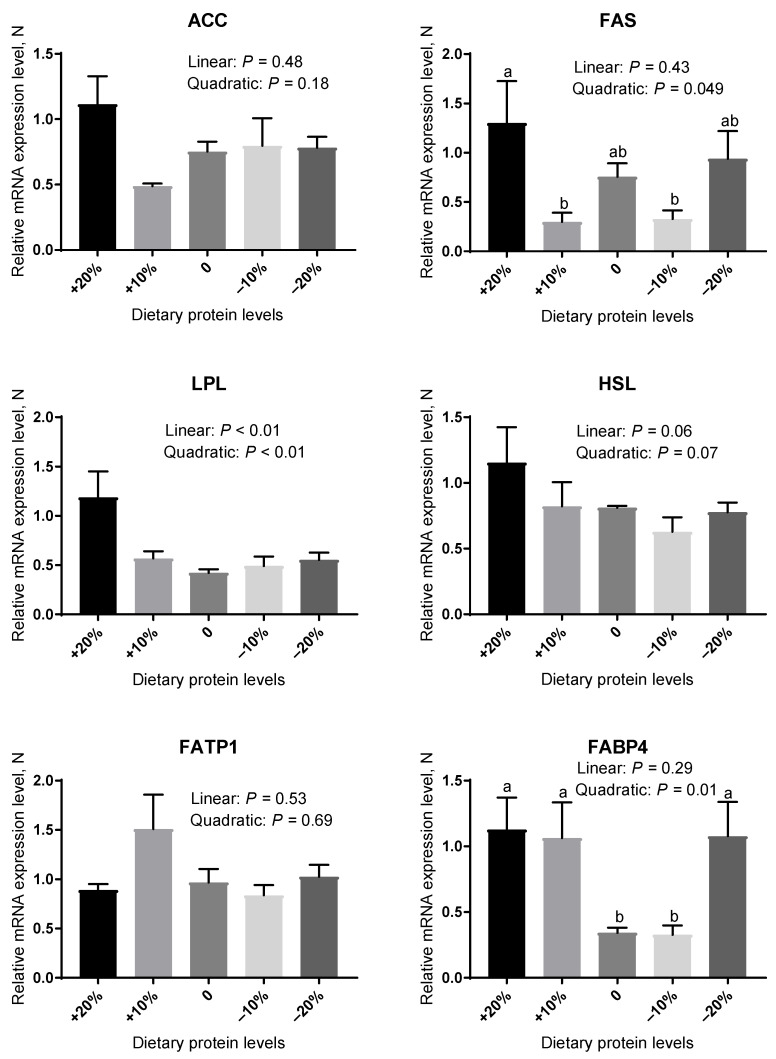
The relative mRNA expression level of genes related to lipid metabolism in the DSA of pigs fed different protein levels. Values are means, with their standard errors represented by vertical bars. ^a,b^ Values (*n* = 6) within a row with different lowercase letters differ significantly (*p* < 0.05) by the Kruskal-Wallis test.

**Figure 2 animals-12-02944-f002:**
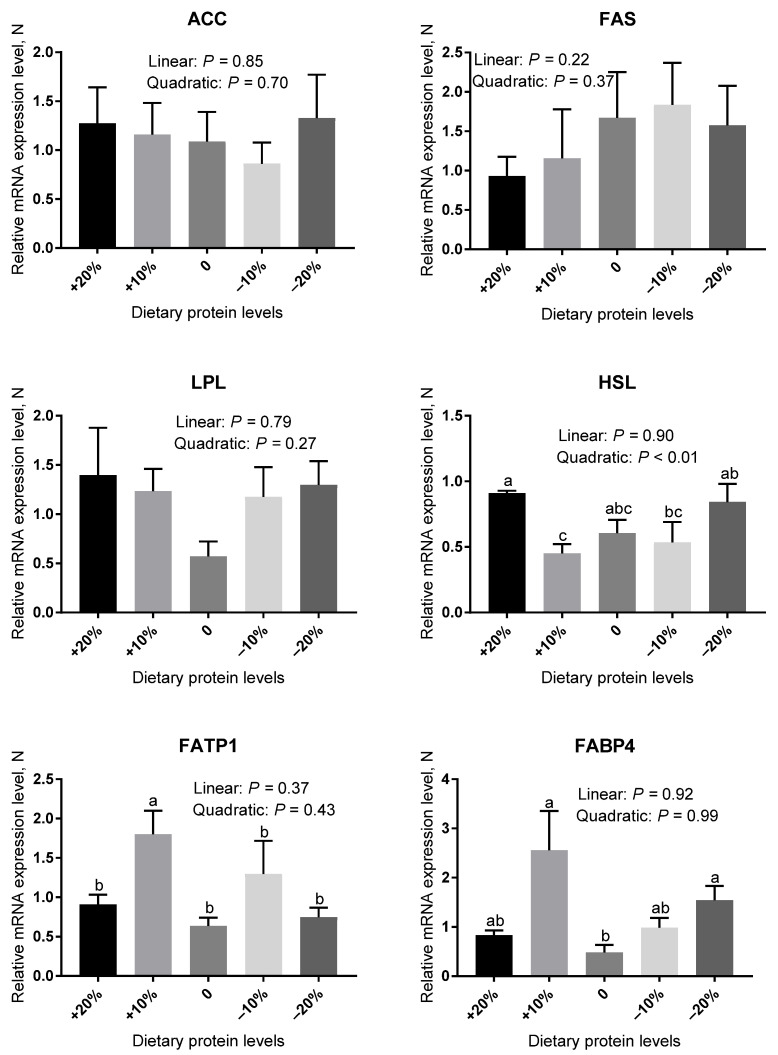
The relative mRNA expression level of genes related to lipid metabolism in the PRA of pigs fed different protein levels. Values are means, with their standard errors represented by vertical bars. ^a,b,c^ Values (*n* = 6) within a row with different lowercase letters differ significantly (*p* < 0.05) by the Kruskal–Wallis test.

**Figure 3 animals-12-02944-f003:**
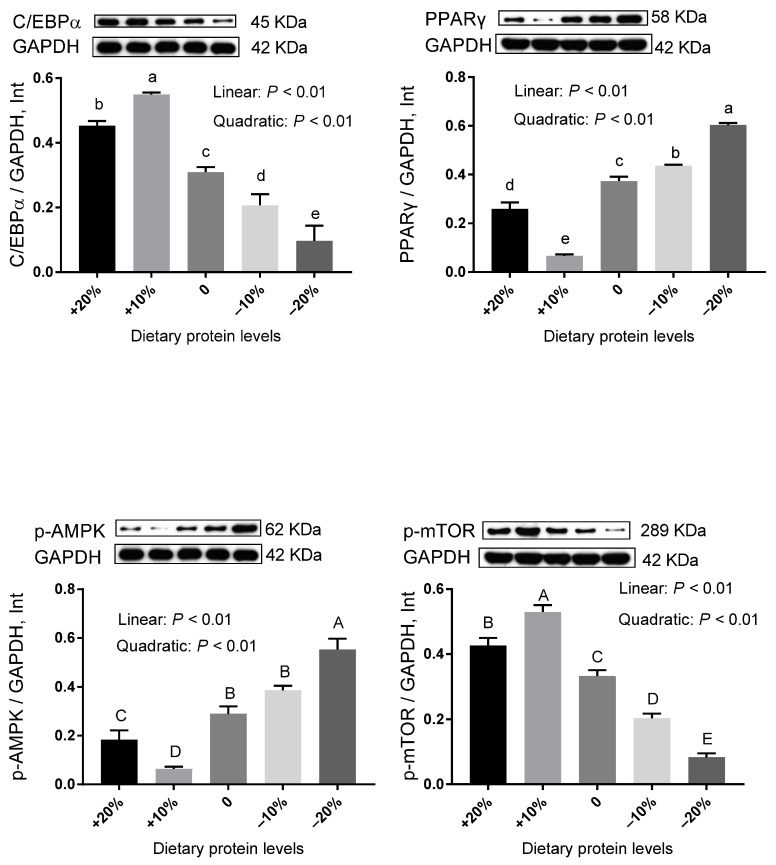
The relative protein expression level of genes related to lipid metabolism in the DSA of pigs fed different protein levels. Values are means, with their standard errors represented by vertical bars. ^a,b,c,d,e^ Values (*n* = 3) within a row with different lowercase letters differ significantly (*p* < 0.05) by the Kruskal–Wallis test. ^A,B,C,D,E^ Values (*n* = 3) within a row with different capital superscripts differ significantly (*p* < 0.05) by Duncan’s multiple comparisons.

**Figure 4 animals-12-02944-f004:**
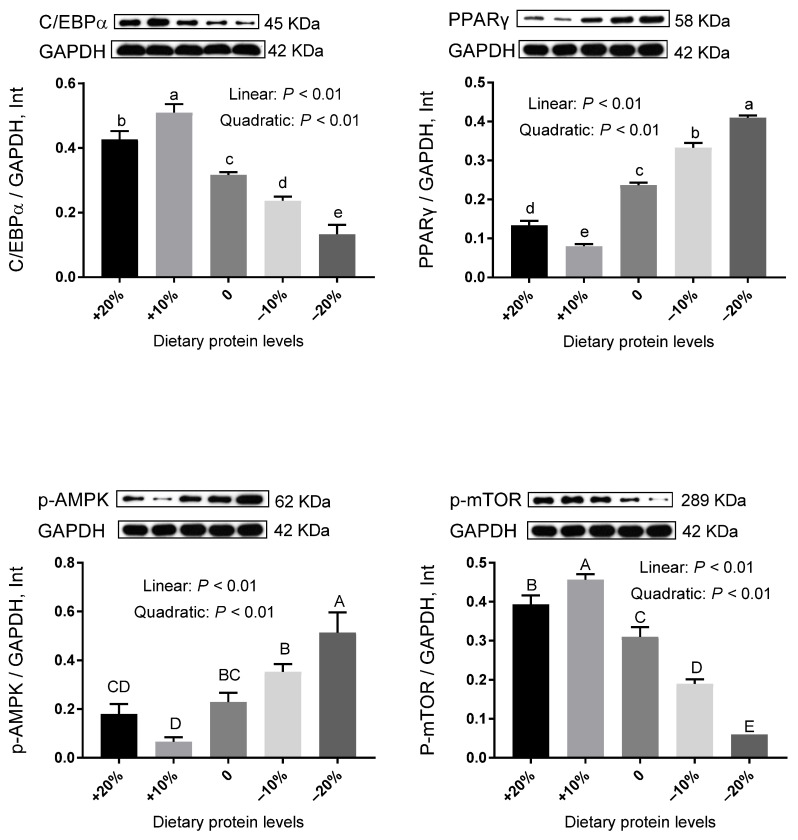
The relative protein expression level of genes related to lipid metabolism in the PRA of pigs fed different protein levels. Values are means, with their standard errors represented by vertical bars. ^a,b,c,d,e^ Values (*n* = 3) within a row with different lowercase letters differ significantly (*p* < 0.05) by the Kruskal–Wallis test. ^A,B,C,D,E^ Values (*n* = 3) within a row with different capital superscripts differ significantly (*p* < 0.05) by Duncan’s multiple comparisons.

**Figure 5 animals-12-02944-f005:**
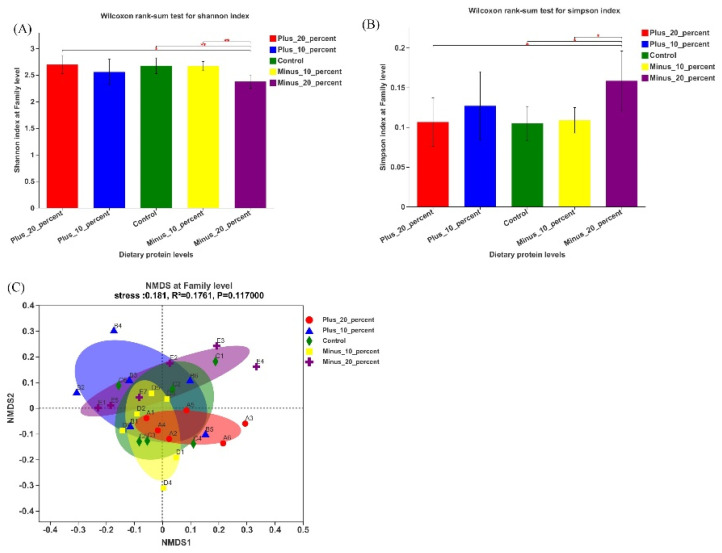
Shannon index plot, Simpson index plot, and NMDS plot for colonic microbiota of pigs fed different protein levels (*n* = 6). (**A**) Shannon index plot of family level. (**B**) Simpson index plot of family level. (**C**) NMDS plot of family level. Plus_20_percent = +20%, Plus_10_percent = +10%, Minus_10_percent = −10%, Minus_20_percent = −20%. Significance levels were marked as * for *p* < 0.05 and ** for *p* < 0.01.

**Table 1 animals-12-02944-t001:** Primers used for real-time quantitative PCR.

Genes ^1^	Primers	Sequences (5′ to 3′)	Product Size, bp
ACC	Forward	AGCAAGGTCGAGACCGAAAG	169
	Reverse	TAAGACCACCGGCGGATAGA	
FAS	Forward	CTACCTTGTGGATCACTGCATAGA	114
	Reverse	GGCGTCTCCTCCAAGTTCTG	
HSL	Forward	CACAAGGGCTGCTTCTACGG	167
	Reverse	AAGCGGCCACTGGTGAAGAG	
LPL	Forward	CTCGTGCTCAGATGCCCTAC	148
	Reverse	GGCAGGGTGAAAGGGATGTT	
FATP1	Forward	ACCACTCCTACCGCATGCAG	78
	Reverse	CCACGATGTTCCCTGCCGAGT	
FABP4	Forward	CAGGAAAGTCAAGAGCACCA	227
	Reverse	TCGGGACAATACATCCAACA	
β-actin	Forward	TGCGGGACATCAAGGAGAAG	216
	Reverse	AGTTGAAGGTGGTCTCGTGG	

^1.^ ACC: acetyl-CoA carboxylase, FAS: fatty acid synthase, HSL: hormone-sensitive lipase, LPL: lipoprotein lipase, FATP1: fatty acid transport protein 1, FABP4: fatty acid binding protein 4.

**Table 2 animals-12-02944-t002:** Blood chemical parameters of lipid metabolism of pigs fed different protein levels.

Item ^1^	Dietary Protein Levels					SEM	*p*-Value		
	+20%	+10%	0	−10%	−20%		ANOVA	Linear	Quadratic
TG, mmol/L	0.31	0.38	0.33	0.40	0.44	0.02	0.30	0.06	0.17
CHOL, mmol/L	2.82	2.31	2.35	2.60	2.52	0.07	0.16	0.51	0.17
LDL-C, mmol/L	1.96	1.62	1.82	1.77	1.50	0.07	0.29	0.12	0.29
HDL-C, mmol/L	1.06	0.78	0.86	1.05	1.10	0.04	0.09	0.30	0.07

^1.^ TG: triglyceride, CHOL: cholesterol, LDL-C: low density lipoprotein cholesterol, HDL-C: high-density lipoprotein cholesterol.

**Table 3 animals-12-02944-t003:** The relative abundances of the top 10 microbial genera (%) in the colonic contents of pigs fed different protein levels.

Item	Dietary Protein Levels					SEM	*p*-value		
	+20%	+10%	0	−10%	−20%		ANOVA	Linear	Quadratic
*Clostridium sensu stricto 1*	0.08	0.13	0.11	0.11	0.12	0.00009	0.53	0.33	0.54
*Treponema*	0.08	0.06	0.06	0.02	0.18	0.00019	0.25	0.19	0.05
*UCG-005*	0.10	0.09	0.05	0.10	0.05	0.00008	0.05	0.17	0.39
*Terrisporobacter*	0.05	0.08	0.09	0.05	0.05	0.00006	0.12	0.91	0.10
*Unclassified f Lachnospiraceae*	0.04 ^c^	0.04 ^c^	0.05 ^bc^	0.07 ^a^	0.07 ^ab^	0.00004	0.02	<0.01	<0.01
*Turicibacter*	0.04	0.06	0.05	0.05	0.07	0.00005	0.18	0.07	0.20
*Lachnospiraceae XPB1014 group*	0.06 ^ab^	0.03 ^bc^	0.02 ^c^	0.05 ^ab^	0.06 ^a^	0.00006	<0.01	0.54	0.01
*Christensenellaceae R-7 group*	0.03 ^b^	0.08 ^a^	0.03 ^b^	0.04 ^ab^	0.05 ^ab^	0.00009	0.04	0.90	0.92
*Lactobacillus*	0.06	0.02	0.04	0.04	0.01	0.00007	0.09	0.13	0.30
*Streptococcus*	0.03	0.02	0.01	0.01	0.03	0.00004	0.21	0.67	0.07

^a,b,c^ Values (*n* = 6) within a row with different lowercase letters differ significantly (*p* < 0.05) by the Kruskal–Wallis test.

## Data Availability

The data presented in this study are available on request from the corresponding author.

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
