# Peer review of "Long-Term Protein Restriction Modulates Lipid Metabolism in White Adipose Tissues and Alters Colonic Microbiota of Shaziling Pigs"

_animals, 2022, doi:10.3390/ani12212944_

Round 1

Reviewer 1 Report

I have read and reviewed your manuscript entitled "Long-Term Protein Restriction Modulates Lipid Metabolism in 2 White Adipose Tissues and Alters Colonic Microbiota of Shaz-3 iling Pigs ". In my opinion, the Authors present valuable data, however, the current form of the paper does not fit for publication in the Animals journal. The list of my comments and suggestions is presented below:

1.       What type of reverse transcription was used? Please add all information about reverse transcription in chapter 2.4.

2.       Chapter 2.4.  This chapter does not follow the MIQE guidelines for publication of  qPCR results (primers concentration, method thermal profile etc.). Have you performed primers validation? I found the information that the real-time PCR was performed as described in Duan et al. 2017, however the sequence of β-actin primers and product size were different. Please explain.

3.       Chapter 2.5. was specified enough, please provided information such as: protein measurement method,   antibodies dilution, blocking method, the size of the sample volume (how many ug exactly). I would like to receive of all of the pictures in full size (i.e. uncropped) blots from all performed experiments that have contributed to the quantitation process.

4.       Please add units to all graphs. Please add protein size to protein bands. What is the meaning of “Protein levels” on the graphs, this is quite confusing.

5.       Supplementary should be also prepared in English. I did not find reference to second supplementary in your manuscript, please provide this information.

6.       The paper needs language improvements.

Author Response

Please, find our explanations in the attached file

Reviewer 2 Report

Obesity is a global concern for humans. Pigs were considered a good model for studying obesity because of their similar anatomy and physiology. In this study, the authors investigate the effects of long-term protein restriction from piglets to finishing pigs by using qPCR and 16s rRNA gene sequencing methods. Results suggested that long-term protein restriction changed the lipid metabolism and colonic microbiota of Shaziling pigs. This study provided a reference for treating obesity by low-protein diets. I suggested accepted this article for publication after revising several minor points. The comments are as follows:

Line 57. I suggest additional background related to the relationship between obesity and bacteria should be added here.

Line 139. Please double-check the chemical formula (ddH2O).

Figure 1-5. The figure should be revised to improve readability. Such as font, text size, and resolution (Figure 5).

Line 236. Language need to be improved to avoid obvious grammatical mistakes.

Line 304. The links between obesity and gut microbiota have been widely discussed in previous studies. The authors found several bacterial genera were changed in different treatment groups. But, sufficient discussion about the changes was lacking here.

Author Response

(The authors gave the same response as above.)

Reviewer 3 Report

Thank you very much for giving me a chance to review "Long-Term Protein Restriction Modulates Lipid Metabolism in White Adipose Tissues and Alters Colonic Microbiota of Shaz-iling Pigs" The study is interesting and topic is hot for several points of view. Materials and methods are comprehensive and the results are clear. The conclusion section is clear and concise. However, be the study is accepted for publication, several points must be clarified by the authors. 

The authors need to identify the novelty of the study and must be incorporated into the introdcution section. The authors have already given several previous works where the reduced protein levels have shown good results. However, how their own work is different from them must be identified and incorporated. 

Also the authors need to clarify why they have chosen the two levels of proteins at their respective levels and how?

In the statistical analysis section, Duncan multiple test is now an old test, the authors need to write the actual and extact test they have applied. The figures are very fade and cannot be seen properly. The authors should try to present them well. 

The conclusion need to be revised. It should be based on the results section and be concize. 

Author Response

(The authors gave the same response as above.)

Round 2

Reviewer 1 Report

I have read your resonse and in my opinion, all time, the current form of the paper does not fit for publication in the Animals journal. The list of my comments and suggestions is presented below:

1.       “Chapter 2.4.  This chapter does not follow the MIQE guidelines for publication of  qPCR results (primers concentration, method thermal profile etc.). Have you performed primers validation? I found the information that the real-time PCR was performed as described in Duan et al. 2017, however the sequence of β-actin primers and product size were different” - I am not satisfied your answer, please explain few thinks. First, you changed the β-actin product size and you have got the same product size like Duan et al. 2017, but you have got another sequence of β-actin primers, please check the sequence of all primer and product size. Second, have you performed primers validation? I have not got the answer for this. Third, you wrote that you used 2 μL cDNA and 0.4 μL each primer, I think it is not a correct statement, because you have to write the concentration of cDNA and primers. The present description does not allow the experiment to be repeated because it does not contain the necessary data.

2.       Chapter 2.5. was specified enough, please provided information such as: protein measurement method,   antibodies dilution, blocking method, the size of the sample volume (how many ug exactly). I would like to receive of all of the pictures in full size (i.e. uncropped) blots from all performed experiments that have contributed to the quantitation process.  - I am not satisfied your answer, please explain few thinks. How many ug of proteins exactly was used? I read that you prepared n=3 for Western Blot, but you send pictures only for n=1 and cropped. I would like to receive of all of the pictures (n=3) of each protein and in full size (i.e. uncropped, with marker).

3.       Please add units to all graphs. Please add protein size to protein bands. What is the meaning of “Protein levels” on the graphs, this is quite confusing. – You wrote me that “They have no units.” But you have to use arbitrary unit.

Author Response

Please, find our explanations in the attached file.
